# Evening Primrose Oil Improves Chemotherapeutic Effects in Human Pancreatic Ductal Adenocarcinoma Cell Lines—A Preclinical Study

**DOI:** 10.3390/ph15040466

**Published:** 2022-04-12

**Authors:** Laura Zeppa, Cristina Aguzzi, Giorgia Versari, Margherita Luongo, Maria Beatrice Morelli, Federica Maggi, Consuelo Amantini, Giorgio Santoni, Oliviero Marinelli, Massimo Nabissi

**Affiliations:** 1School of Pharmacy, University of Camerino, 62032 Camerino, MC, Italy; laura.zeppa@unicam.it (L.Z.); cristina.aguzzi@unicam.it (C.A.); giorgia.versari@studenti.unicam.it (G.V.); mariabeatrice.morelli@unicam.it (M.B.M.); giorgio.santoni@unicam.it (G.S.); 2Integrative Therapy Discovery Lab, University of Camerino, 62032 Camerino, MC, Italy; 3“Maria Guarino” Foundation—AMOR No Profit Association, 80078 Pozzuoli, NA, Italy; margherita.luongo@aslnapoli2nord.it; 4School of Bioscience and Veterinary Medicine, University of Camerino, 62032 Camerino, MC, Italy; federica.maggi@unicam.it (F.M.); consuelo.amantini@unicam.it (C.A.)

**Keywords:** pancreatic cancer, human pancreatic ductal adenocarcinoma, evening primrose oil, *Oenothera biennis*, chemoresistance, paclitaxel chemoresistance, cytotoxicity

## Abstract

Evening Primrose oil (EPO), obtained from the seeds of Evening Primrose (*Oenothera* L.), is largely used as a dietary supplement, especially after cancer diagnosis. Human pancreatic ductal adenocarcinoma (PDAC) is an aggressive disease correlated with poor clinical prognosis and a very low response rate to common chemotherapy. The aim of this work was to study the potential ability of EPO to improve the effects of chemotherapeutic drugs in PANC-1 and MIAPaCa-2 cell lines. Cytotoxicity, cell death, reactive oxygen species (ROS) production and EPO anticancer activity associated with the main chemotherapeutic drugs commonly used in therapy were investigated. Results showed that EPO reduced PDAC cell viability and increased paclitaxel efficacy. This evidence suggests that EPO may be used as a potential supplement to increase chemotherapeutic efficacy in PDAC therapy.

## 1. Introduction

Research of natural derivatives to be used as an adjuvant in cancer therapy is of current interest, and several studies have reported the potential anticancer activity of compounds derived from medicinal plants [1,2,3,4,5]. Different studies indicate that cancer patients make dietary changes after cancer diagnosis [6], as reported for example by the DietCompLyf study for breast cancer cases. Among the main supplements, Evening Primrose oil (EPO) is largely used, and the percentage of consumers significantly increases after diagnosis [7]. 

Evening Primrose (*Oenothera* L.) belongs to the Onagraceae family, and plants from the genus *Oenothera biennis* are the most numerous and have been studied extensively. EPO, obtained from the seeds, is used as a dietary supplement [8,9,10], and it is composed of a 98% mixture of triacylglycerols and a 1–2% non-saponifiable fraction made of 53.16% sterols. Among the triacylglycerols, 70–74% are composed of linoleic acid (LA) and another 8–10% of gamma-linolenic acid (GLA), which are essential polyunsaturated fatty acids (PUFAs) belonging to the omega-6 acids family [8,9,10]. The beneficial effects of EPO consumption on the skin, in different autoimmune diseases, in premenstrual syndrome and in reducing low-density lipoprotein (LDL) levels are associated with the elevated presence of PUFAs, but other effects are also associated with the presence of sterols [8,9,10].

Cancer is a leading cause of death worldwide, and pancreatic cancer is ranked as the 14th most common cancer and the 7th highest cause of cancer-related mortality. Pancreatic cancer, including pancreatic ductal adenocarcinoma (PDAC), is one of the most aggressive and malignant solid cancers, with a 5-year survival of approximately 5–9%. PDAC is an infiltrating neoplasm with glandular differentiation derived from the pancreatic ductal tree [11,12]. It has demonstrated a family genetic predisposition, but precursor lesions within pancreatic tissue and somatic mutations of *KRAS* oncogene and *CDKN2A*, *TP53* and *SMAD4* suppressor genes are also implicated in PDAC pathogenesis [13,14]. The main therapeutic approach for PDAC is surgical resection with adjuvant chemotherapy, but surgery is not always possible, especially in the case of metastases, and, moreover, PDAC response to chemotherapeutic drugs is very low. Given that PDAC is still considered an incurable cancer, new drugs and adjuvant supplementation are necessary to reduce mortality and improve therapeutic outcomes [4].

There has been no preclinical evidence regarding the potential anticancer effects of EPO up to now. In this study, we evaluated the capability of EPO to promote cytotoxicity in human PDAC cell lines and improve chemotherapeutic drug activity, in order to support the use of EPO supplementation in the diet of PDAC patients.

## 2. Results

### 2.1. EPO Reduces PANC-1 and MIAPaCa-2 Cell Viability

EPO cytotoxicity was investigated in PANC-1 and MIAPaCa-2 cells by 3-(4,5-dimethylthiazol-2-yl)-2,5-diphenyltetrazolium bromide (MTT) assay. Cells were treated with different doses of EPO (from 0.024 up to 50 μL/mL). After 72 h, results showed a reduction in cancer cell viability in a dose-dependent manner, with an IC_50_ of 1.37 ± 0.06 μL/mL for PANC-1 and 0.89 ± 0.02 μL/mL for MIAPaCa-2 (Figure 1A). MIAPaCa-2 cells were more sensitive to treatment than PANC-1. 

The EPO cytotoxic effect was also evaluated in the noncancerous human keratinocyte HaCaT cell line (Figure 1B). Cells were treated with the same doses of EPO used in PANC-1 and MIAPaCa-2 (0.024–50 μL/mL). After 72 h, doses higher than 6.25 μL/mL reduced HaCaT cell viability. This result shows that EPO is more cytotoxic on cancer cells compared to normal cells.

### 2.2. EPO Induces Cell Death in PANC-1 and MIAPaCa-2 Cell Lines

Propidium iodide (PI) staining was used to evaluate cancer cell death. Cells were treated with the lowest cytotoxic doses of EPO (0.39 μL/mL and 0.78 μL/mL), and after 48 h, an increase in cells undergoing cell death was observed in both cell lines. In PANC-1, cell death was induced predominantly with 0.78 μL/mL and in MIAPaCa-2 with 0.39 μL/mL EPO (Figure 2).

To further confirm the EPO-dependent cell death, Western blot analysis was performed, and H2AX presence, as a marker of DNA damage, was evaluated in PANC-1 and MIAPaCa-2. PDAC cells were treated with 0.39 μL/mL and 0.78 μL/mL EPO for 48 h. Results showed a significant increase in phospho-histone (H2AX) protein expression in MIAPaCa-2 with both doses and only with 0.78 μL/mL in PANC-1, confirming a lower sensitivity of this cell line (Figure 3).

### 2.3. EPO Induces ROS Production in PANC-1 and MIAPaCa-2 Cell Lines

Since EPO induced cell death, the involvement of intracellular ROS production was evaluated. PANC-1 and MIAPaCa-2 cells were treated with EPO at 0.39 μL/mL and 0.78 μL/mL, and ROS production was analyzed by cytofluorimetric analysis after 2 and 4 h. Results showed that both EPO doses increased ROS production in PANC-1 cells, while for MIAPaCa-2, the increase in ROS started with 0.78 μL/mL EPO. ROS formation was confirmed through the pretreatment of PANC-1 and MIAPaCa-2 cells with N-acetyl-l-cysteine (NAC), an inhibitor of ROS production (Figure 4).

### 2.4. EPO Potentiates Paclitaxel Efficacy in PANC-1 and MIAPaCa-2 Cell Lines

The EPO cytotoxic effect was evaluated in combination with PTX and gemcitabine (GEM), the main chemotherapeutic drugs currently used in PDAC therapy [13]. PANC-1 and MIAPaCa-2 cells were treated with different doses of EPO (from 0.095 up to 0.39 μL/mL), GEM (from 12.5 to 50 µg/mL) and PTX (from 0.75 to 3 µg/mL) alone or in combination for 72 h.

Data showed that EPO was not able to increase the cytotoxicity of GEM (Figure 5), and according to isobologram (data not shown), there are no synergistic/additive effects.

On the contrary, some combinations of EPO and PTX showed synergistic or additive effects (Figure 6). Indeed, combinations with 3 µg/mL PTX and the three EPO doses showed a synergistic effect, as suggested by the combination index (CI) values (0.16979, 0.1869 and 0.20108 for PANC-1, 0.1009, 0.14078 and 0.19799 for MIAPaCa-2). Furthermore, two synergistic combinations with 1.5 µg/mL PTX and 0.19 μL/mL or 0.39 μL/mL EPO (CI = 0.78758 and CI = 0.84974, respectively) and an additive effect with 0.75 µg/mL PTX and 0.19 μL/mL EPO (CI = 1.0412) were obtained in PANC-1 cells.

### 2.5. EPO Influences pERK/ERK Protein Levels and Activation 

It is demonstrated that the Ras-ERK pathway contributes to oncogenic processes, leading to cancer progression [15] and reducing the response to chemotherapeutic drugs [16]. Therefore, the expression of ERK protein and its active phosphorylated form, pERK, was evaluated by Western blot analysis after treatment with 0.39 μL/mL and 0.78 μL/mL EPO for 12 and 24 h. In PANC-1 cells, data evidenced a slight reduction of pERK starting from 12 h, followed by a significant decrease in total ERK after 24 h with 0.78 μL/mL EPO (Figure 7).

In MIAPaCa-2 cells, a reduction in pERK was detected after 24 h of treatment with 0.78 μL/mL EPO, while a significant reduction in total ERK was observed after 24 h of incubation with both doses of EPO (Figure 8). 

## 3. Discussion

Recently, plant derivatives have been extensively studied to evaluate their ability to improve the therapy of many chronic diseases and cancer [17]. Therefore, we evaluated the effect of EPO, obtained from Evening Primrose seeds, which are rich in GLA and linoleic acid. While there are many studies about the anticancer effect of GLA [18,19], data on the anticancer effect of EPO are few.

EPO has been reported to reduce many inflammatory conditions, such as skin disorders, atopic dermatitis and rheumatoid arthritis, and its activity has also been demonstrated in diabetes and premenstrual syndrome [8,20,21]. Furthermore, some evidence has shown its potential efficacy in reducing side effects associated with chemotherapy. Indeed, in an in vivo model, EPO pretreatment reduced cyclophosphamide-induced hepatic and pancreatic toxicity, ameliorating biochemical parameters and histopathological alterations [20]. In addition, it reduced skin reaction induced by bortezomib injection in multiple myeloma patients [22].

We evidenced that EPO reduced PANC-1 and MIAPaCa-2 cell viability in a dose-dependent manner after 72 h of treatment. Similarly, different extracts obtained from another variety of Evening Primrose induced a reduction in cancer cell viability. An extract obtained from the defatted seeds of *Oenothera paradoxa* reduced the cell viability and invasiveness of malignant pleural mesothelioma and induced apoptosis in Caco-2 cells [17,23], while phytosterols isolated from EPO and its main components (β-sitosterol and campesterol) decreased the proliferation of human colon adenocarcinoma cell [24]. Then, our results showed that EPO induced cell death in PDAC cells, confirmed by an increase in PI-positive cells and H2AX protein expression. It was demonstrated that Evening Primrose seed extract induces apoptosis in Ehrlich ascites tumor cells [21] and cell death in HT-29 cells [24]. Moreover, we demonstrated that cancer cell death was more evident in MIAPaCa-2 cells, as shown by the lower IC_50_ value obtained from the cell viability assay.

To examine the mechanism of EPO-induced cell death, we evaluated if EPO enhances ROS production. ROS are often associated with cell damage and, in cancer, induce cell death, given that cancer cells are more sensitive to oxidative stress [25,26]. Our data showed that EPO induced ROS production in both cell lines.

Since the aim of integrated therapy is to improve the efficacy of common chemotherapy, we investigated the EPO effect in combination with PTX and GEM. On the one hand, EPO showed a synergistic and/or additive effect in combination with some PTX doses. In particular, synergism was evident with the higher dose of PTX after 72 h of treatment on PANC-1 and MIAPaCa-2. On the other hand, EPO was unable to increase the efficacy of the GEM. Among the main pathways that mediate the response to chemotherapeutic drugs, RAS/RAF/MEK/ERK are interesting targets for cancer therapy [27]. Indeed, ERK is involved in many biological processes, including cancer progression and resistance [28].

There are no previous data on the combined effect between EPO and chemotherapeutic drugs. Studies have reported that the combination of trametinib, a selective inhibitor of MEK1/2 kinase activity, and GEM does not lead to an efficient clinical response in PDAC phase I clinical trials. On the contrary, in non-small-cell lung cancer, selumetinib, another MEK inhibitor, demonstrated some benefits combined with docetaxel. This supports the potential synergistic effect of MEK inhibitors with taxane derivatives [16]. Similarly, we demonstrated that EPO reduced ERK phosphorylation after 24 h and total ERK levels in both cell lines, and this may justify the increase in PTX efficacy.

Furthermore, EPO is rich in PUFAs, and it was demonstrated that they are able to affect MEK/ERK pathway. Indeed, dietary long-chain n-3 PUFAs are able to inhibit the activation of these pro-survival pathways in in vitro and in vivo breast cancer models [29], supporting our evidence.

## 4. Materials and Methods

### 4.1. Cell Lines

Human pancreatic ductal adenocarcinoma (PANC-1 and MIAPaCa-2) cell lines were obtained from Sigma Aldrich (Milan, Italy), and immortalized human keratinocytes cell line (HaCaT) furnished by IFOM (Institute of Molecular Oncology, Rome, Italy) was maintained in DMEM glucose ^high^ medium (EuroClone, Milan, Italy) supplemented with 100 IU/mL penicillin, 100 mg streptomycin, 10% fetal bovine serum (FBS), 1 mM sodium pyruvate and 2 mM L glutamine. Cell lines were cultured at 37 °C with 5% CO_2_ and 95% humidity.

### 4.2. Reagents

Evening Primrose oil (EPO), derived from *Oenothera biennis*, was purchased (Product No. PHR 2978; Sigma Aldrich, Milan, Italy) as pharmaceutical standard, certified reference material (composition described in Table 1) and diluted in 1:2 DMSO, and it was freshly prepared for each experiment. Gemcitabine (GEM; 50 mg/mL) and paclitaxel (PTX; 6 mg/mL) were purchased by Sigma Aldrich (Milan, Italy), solubilized in water and stored at −20 °C.

### 4.3. MTT Assay

In 96-well plates, cells were seeded at a concentration of 3 × 10^4^ cells/mL in a final volume of 100 μL/well. After 24 h of incubation, treatments were administered in six replicates for each. After 72 h, cell viability was investigated by adding 0.8 mg/mL of MTT obtained from Sigma Aldrich (Milan, Italy) to the media, according to the protocol previously described [4].

### 4.4. Cell Death Assay

To evaluate cell death on PANC-1 and MIAPaCa-2 cells, PI staining was used and analyzed by FACScan. Cells were plated at a density of 5 × 10^4^ cells/mL, and two doses of EPO were added for 48 h. After treatment, cells were stained with 20 μg/mL PI for 10 min at room temperature and washed. CellQuest software version 3.0 (BD Biosciences, San Jose, CA, USA) was used to determine the percentage of positive cells. All experiments were repeated three times.

### 4.5. Western Blot Analyses

PANC-1 and MIAPaCa-2 cell lines, untreated or treated with EPO for 12 and 24 h, were lysed with lysis buffer, composed as previously described [4]. Lysates were separated on SDS polyacrylamide gel, transferred onto Hybond-C extra membranes (GE Healthcare, Chicago, IL, USA), blocked with 5% of bovine serum albumin (BSA) in PBS-Tween 20, immunoblotted with rabbit anti-phospho-histone H2AX (Ser139) (1:1000, #9718, Cell Signaling Technology, Danvers, MA, USA), mouse anti-pERK (1:2.000, Cell Signaling Technology, Danvers, MA, USA) and rabbit anti-ERK (1:1.000, Cell Signaling Technology, Danvers, MA, USA) Abs overnight. Mouse anti-glyceraldehyde-3-phosphate dehydrogenase (GAPDH, 1:1000, Santa Cruz, CA, USA) Ab was incubated for one hour, and then all were incubated with their respective HRP-conjugated anti-rabbit secondary Abs (1:5.000, Jackson ImmunoResearch Europe Ltd., Cambridge, UK) and anti-mouse secondary Abs (1:2.000, Cell Signaling Technology, Danvers, MA, USA) for one hour. The detection was performed using the LiteAblot ^®^PLUS or the LiteAblot ^®^TURBO (EuroClone, Milano, Italy) kits. Densitometric analysis was carried out using the Quantity One software (BioRad, Hercules, CA, USA). H2AX and ERK densitometry values were normalized to GAPDH used as loading control and compared to vehicle, while pERK densitometry value was normalized to ERK. Densitometric values shown are the mean ± SD of three separate experiments.

### 4.6. Reactive Oxygen Species (ROS) Production

In order to assess the oxidative stress, a fluorescent dichlorodihydrofluorescein diacetate (DCFDA) probe was used. Briefly, 5 × 10^4^ cells, seeded on a 12-well plate and treated with two EPO doses, were incubated with 20 μM DCFDA (Life Technologies Italia, Monza, Italy) 20 min prior to the time point. N-acetylcysteine (NAC) (10 mM) with preincubation of 3 h was used as control. The intensity of the fluorescence was analyzed using FACScan and CellQuest software version 3.0 (BD Biosciences, San Jose, CA, USA).

### 4.7. Statistical Analysis

Mean and standard deviation are the result of three independent experiments. The statistical significance was determined by one-way ANOVA test with Tukey’s multiple comparisons post-hoc test. Synergistic and additive effects were calculated by the Chou–Talalay method as previously described [4]. CompuSyn Software 3.0.1 version (ComboSyn, Inc., Paramus, NJ, USA, 2007) was used for automatically determining synergism and antagonism. Statistical analysis of IC_50_ was performed by Prism 5.01 (Graph Pad Software, San Diego, CA, USA).

## 5. Conclusions

In conclusion, EPO significantly reduced PDAC cell viability, inducing cell death and improving PTX efficacy. This evidence suggests its potential use as an adjuvant in PDAC therapy. Further studies should be performed to elucidate other molecular mechanisms involved in EPO-induced anticancer activity. Since these data were performed in cell lines, in vivo preclinical data could be useful to confirm the in vitro findings. 

## Figures and Tables

**Figure 1 pharmaceuticals-15-00466-f001:**
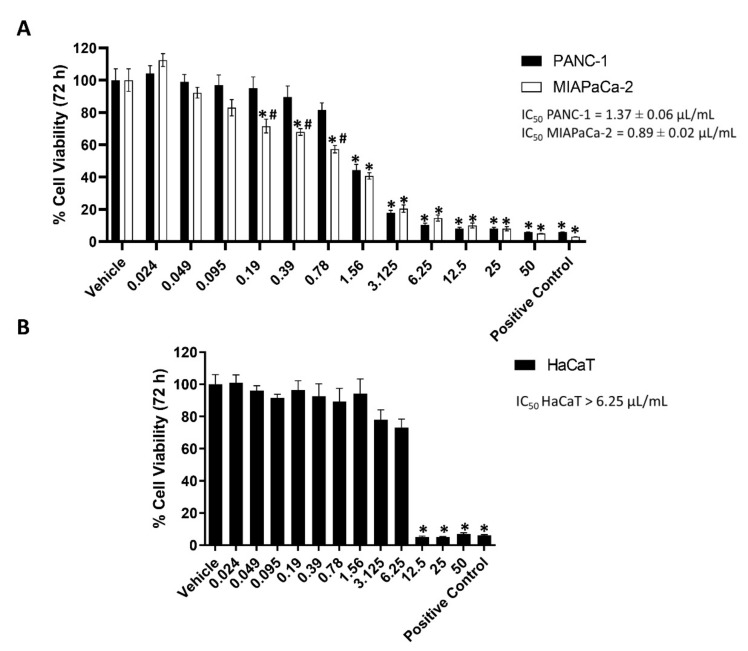
(**A**) PANC-1 and MIAPaCa-2 cell viability after treatment with different concentrations of EPO (μL/mL) for 72 h. Vehicle (dimethyl sulfoxide (DMSO)) was used as negative control and paclitaxel (PTX) 23.9 μg/mL as positive control. Data shown are expressed as mean ± SD of three separate experiments. * *p* < 0.05 treated PANC-1 or MIAPaCa-2 vs. vehicle, ^#^
*p* < 0.05 treated PANC-1 vs. treated MIAPaCa-2. (**B**) HaCaT cell viability after treatment with different concentrations of EPO (μL/mL) for 72 h. Vehicle (DMSO) was used as negative control and 23.9 μg/mL PTX as positive control. Data shown are expressed as mean ± SD of three separate experiments. * *p* < 0.05 treated HaCaT vs. vehicle.

**Figure 2 pharmaceuticals-15-00466-f002:**
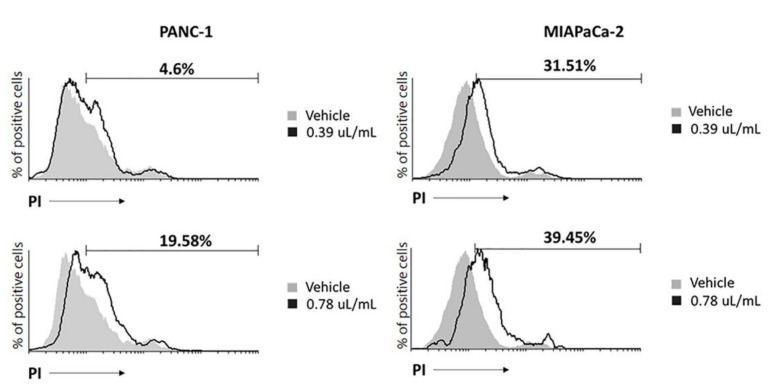
EPO induces cell death in PANC-1 and MIAPaCa-2 after 48 h. Data represent the percentage of PI-positive cells and are representative of one of three separate experiments.

**Figure 3 pharmaceuticals-15-00466-f003:**
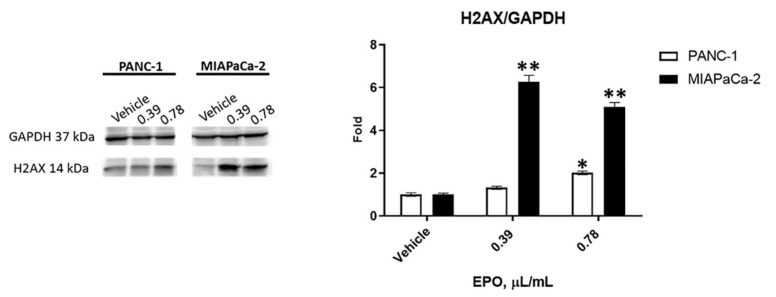
H2AX densitometric values were normalized to glyceraldehyde-3-phosphate dehydrogenase (GAPDH) used as loading control. Densitometric values shown are the mean ± SD of three separate experiments. * *p* < 0.05, ** *p* < 0.01, treated PANC-1 or MIAPaCa-2 vs. vehicle.

**Figure 4 pharmaceuticals-15-00466-f004:**
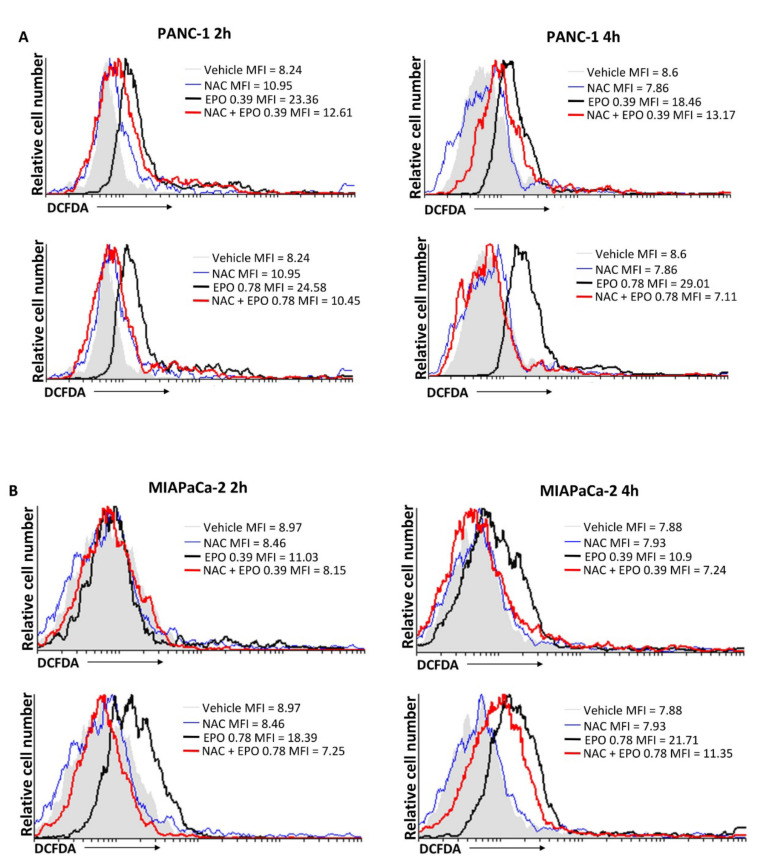
EPO effect on ROS production in (**A**) PANC-1 and (**B**) MIAPaCa-2 cells after 2 and 4 h of treatment. Results are expressed as the mean fluorescence intensity (MFI).

**Figure 5 pharmaceuticals-15-00466-f005:**
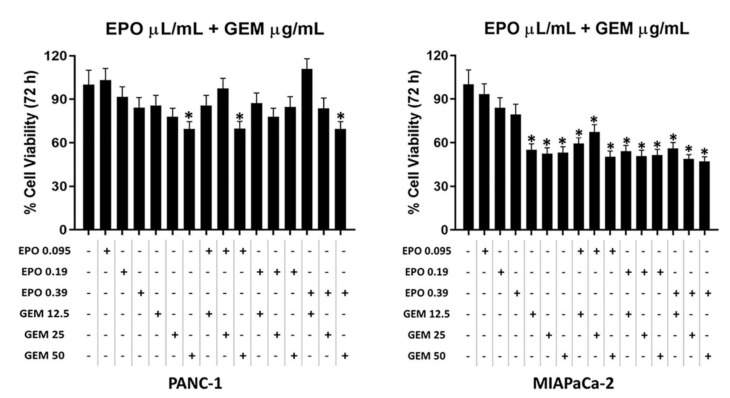
Evaluation of the effect of GEM and EPO combinations on PANC-1 and MIAPaCa-2 cell viability. Data shown are expressed as the mean ± SD of three separate experiments. * *p* < 0.05 treated PANC-1 or MIAPaCa-2 vs. vehicle.

**Figure 6 pharmaceuticals-15-00466-f006:**
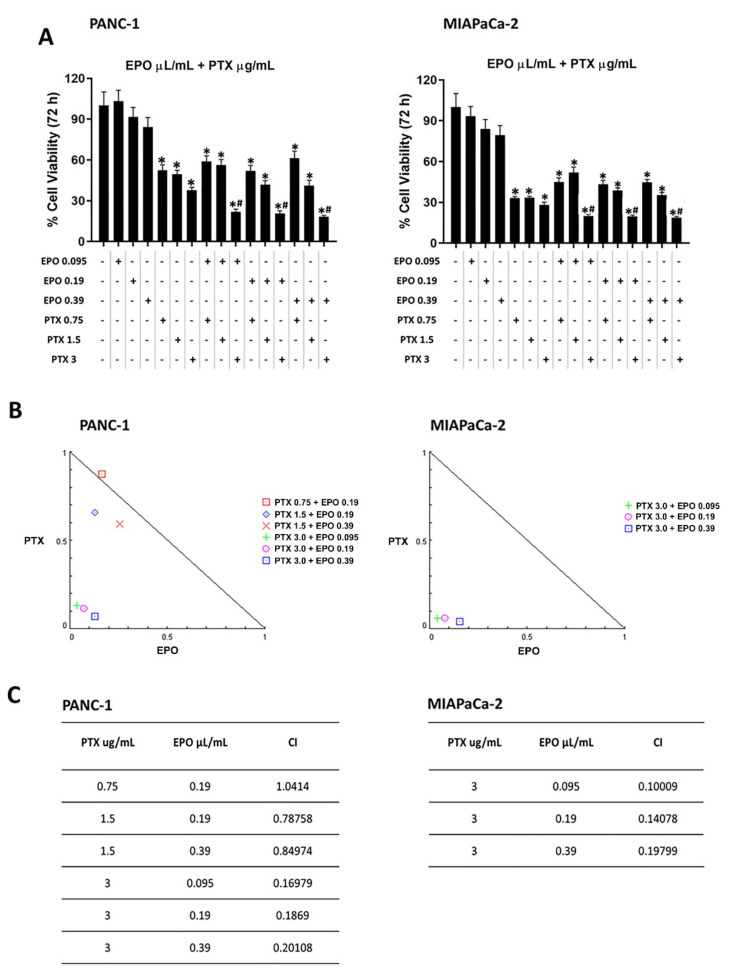
Evaluation of the effect of PTX and EPO combinations in PANC-1 and MIAPaCa-2 cells. (**A**) PANC-1 and MIAPaCa-2 cell viability after treatment with different combinations of PTX and EPO. Data shown are expressed as the mean ± SD of three separate experiments. * *p* < 0.05 treated PANC-1 or MIAPaCa-2 vs. vehicle, ^#^
*p* < 0.05 treated PANC-1 or MIAPaCa-2 vs. PTX alone. (**B**) Isobologram plots for combination treatments of PTX and EPO in PANC-1 and MIAPaCa-2 cell lines. Lower left of the hypotenuse, synergism; on the hypotenuse, additive effect; upper right, antagonism. Synergistic activity of PTX-EPO was calculated by CompuSyn Software. (**C**) CI values for PANC-1 and MIAPaCa-2 cell lines.

**Figure 7 pharmaceuticals-15-00466-f007:**
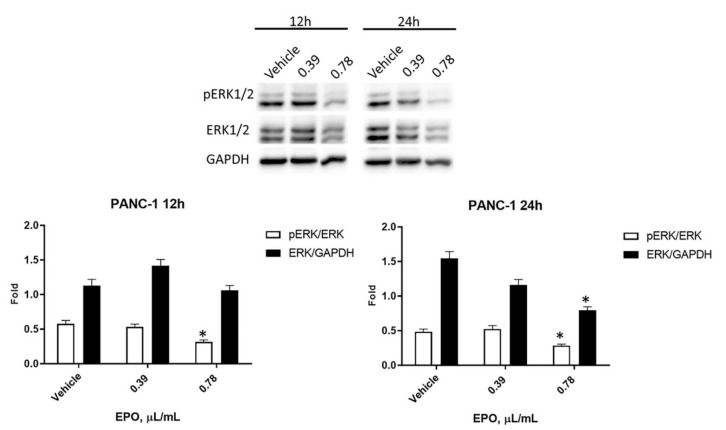
Evaluation of EPO effect in PANC-1. Western blot analysis of pERK and ERK levels in PANC-1 cells after 12 and 24 h treatment with EPO. ERK protein expression was normalized to GAPDH protein expression used as loading control; pERK protein expression was normalized to ERK protein expression. Densitometric values shown are the mean ± SD of three separate experiments. * *p* < 0.05 treated vs. vehicle cells.

**Figure 8 pharmaceuticals-15-00466-f008:**
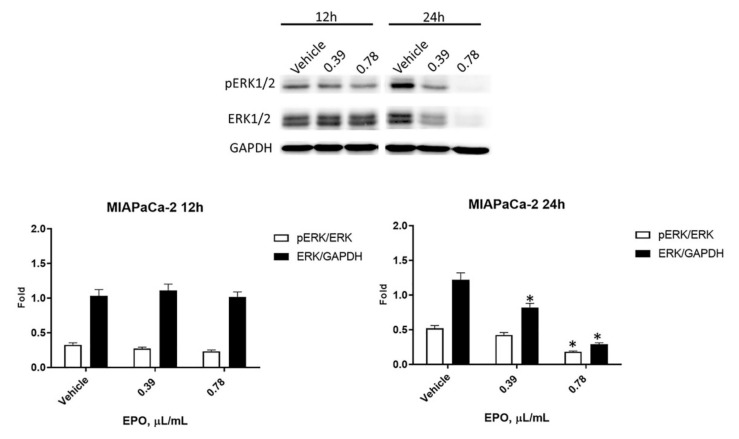
Evaluation of EPO effect in MIAPaCa-2. Western blot analysis of pERK and ERK protein levels in MIAPaCa-2 cells after 12 and 24 h of treatment with EPO. ERK protein expression was normalized to GAPDH protein expression used as loading control; pERK protein expression was normalized to ERK protein expression. Densitometric values shown are the mean ± SD of three separate experiments. * *p* < 0.05 treated vs. vehicle cells.

**Table 1 pharmaceuticals-15-00466-t001:** EPO major component, as described by the manufacturer. United States Pharmacopeia (USP).

Fatty Acid Methyl Esters (FAME)	USP Comp %
Methyl palmitate	6.00
Methyl stearate	1.91
Methyl oleate	6.93
Methyl linoleate	74.18
Methyl-y-linolenate	9.98
Methyl-o-linolenate	0.22
Methyl arachidate	0.49
Methyl eicosenoate	0.16

## Data Availability

Data is contained within the article.

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
