# Peer review of "Evening Primrose Oil Improves Chemotherapeutic Effects in Human Pancreatic Ductal Adenocarcinoma Cell Lines—A Preclinical Study"

_pharmaceuticals, 2022, doi:10.3390/ph15040466_

Round 1

Reviewer 1 Report

The authors prepared another work based on the possibility of using plant-based supplements supporting the treatment of pancreatic cancer.
They chose one of the popular Evening Primrose Oil (EPO) supplements for analysis, demonstrating its usefulness in reducing the viability of cancer lines, especially in combination with paclitaxel.
However, the work lacks several important elements. First of all, there is no information about carrying out controls on non-cancerous cell lines of the pancreas and / or other human cell lines. Why did the authors focus only on  a pharmaceutical standard of EPO obtained from a chemical company? If patients with pancreatic cancer introduce Evening Primrose Oil into their diet, then at least attempts should be made with supplements used by patients, because their effect may be modified by the presence of other substances in the preparations used.
The article is not suitable for publication in its current form. 

Author Response

REVIEWER 1

The authors prepared another work based on the possibility of using plant-based supplements supporting the treatment of pancreatic cancer. They chose one of the popular Evening Primrose Oil (EPO) supplements for analysis, demonstrating its usefulness in reducing the viability of cancer lines, especially in combination with paclitaxel.
However, the work lacks several important elements. First of all, there is no information about carrying out controls on non-cancerous cell lines of the pancreas and / or other human cell lines. Why did the authors focus only on a pharmaceutical standard of EPO obtained from a chemical company? If patients with pancreatic cancer introduce Evening Primrose Oil into their diet, then at least attempts should be made with supplements used by patients, because their effect may be modified by the presence of other substances in the preparations used.
The article is not suitable for publication in its current form. 

RESPONSE

Thanks to the reviewer for the comments.

  1. Regarding evaluation of EPO in non-cancerous human cells, we performed a cytotoxicity assay in normal human keratinocytes (HaCaT). The results show as EPO induced a lower cytotoxic effect in HaCat compared to PDAC cell lines. The result is reported in Fig.1 b.
  2. We used a pharmaceutical standard because regarding Evening Primrose oil supplements, in the market there are different commercial formulations that differ in 2.5-5 % in the composition. The use of commercial EPO supplements would have required the analysis of different commercial formulations.

Reviewer 2 Report

  1. Whether it is Figure 1 or Table 1?
  2. What is FAME in the Table?
  3. The chemical composition of EPO is not clear.
  4. Alone EPO or its derivative should have been screened for activity so that the comparison would have been easier.
  5. Therefore, authors are asked to modify the manuscript as per the suggestions and resubmit for consideration.

Author Response

Thanks for the comments.

  1. Whether it is Figure 1 or Table 1?

   We apologize for the mistake, the sentence “EPO composition was reported (Supplementary Figure 1)” was deleted.   

2. What is FAME in the Table?

FAME – Fatty Acid Methil Esters, as reported by the manufacturer.

  1. The chemical composition of EPO is not clear.

We used a pharmaceutical standard, which composition is reported in Table 1, as described by the manufacturer.

  1. Alone EPO or its derivative should have been screened for activity so that the comparison would have been easier.

In this work, we focused on the synergistic properties of EPO with paclitaxel and Gemcitabine, since as reported in the multicenter study (Ref.7), EPO is one of the largely used supplements after cancer diagnosis.  

Reviewer 3 Report

  1. Figure 1: PANC-1 and MIAPaCa-2 cell viability should be providing negative control and positive control data.
  2. The cellular study in 96-well plates, the sample size is suitable?
  3. The safety evidences of EPO can be provided.
  4. In the clinical, PTX made the toxicity and side effect. Authors can provide improving evidence would be batter.
  5. Section: 4.7 The statistics of post hoc should be addressed.

Author Response

Thanks for the comments.

  1. Figure 1: PANC-1 and MIAPaCa-2 cell viability should be providing negative control and positive control data.

In cytotoxic assay we have used vehicle (DMSO) as negative control while Paclitaxel (23.9 μg/mL) was added as positive control as previously evidenced (Ref. 4). Figure 1a has been modified as requested.

  1. The cellular study in 96-well plates, the sample size is suitable?

MTT assay is widely used for drug screening and to provide cytotoxicity data and approved by National Cancer Institute (NCI). MTT protocol assay is based on 96 wells plate test, with at least three replicates for dose.

  1. The safety evidences of EPO can be provided.

Regarding the evaluation of EPO in non-cancerous human cells, we performed a cytotoxicity assay in normal human keratinocytes (HaCaT). The results showed as EPO induced a lower cytotoxic effect in HaCaT compared to PDAC cell lines. Result is reported in Fig.1 b.

  1. In the clinical, PTX made the toxicity and side effect. Authors can provide improving evidence would be batter.

In vitro data evidenced that the combination of EPO with PTX induced a synergistic effect. This data showed that similar cytotoxicity can be provided using EPO in combination with lower doses of PTX compared to higher PTX doses. So, the use of lower doses of PTX should be very important to reduce PTX side effects without reducing its anticancer activity.

  1. Section: 4.7 The statistics of post hoc should be addressed.

We have performed a statistical analysis by One-Way ANOVA test with a Tukey’s multiple comparisons post-hoc test. We added in statistical analysis method in Materials and Methods.

Reviewer 4 Report

This manuscript is a revised version and appears to have addressed the points raised by previous reviewers. There are a few minor points that require checking:

1) panel B in fig. 6, axes and points should be labeled (e.g., which doses correspond to point 1? and so on; there are less than 9 points on the graphs, where are the missing points?), and the authors should comment the results of the analysis.

2) what are the differences between PANC-1 and MIAPaCa-2 cell lines?

3) English language should be edited by a native English speaker

4) All non-standard abbreviations should be defined the first time they are used

Author Response

Review response

  • panel B in fig. 6, axes and points should be labeled (e.g., which doses correspond to point 1? and so on; there are less than 9 points on the graphs, where are the missing points?), and the authors should comment the results of the analysis.

Thank you for these appropriate comments. The Figure 6 was modified as requested, and result comments described. Regarding the 9 points, the software CompuSyn generates the isobologram excluding the C.I. values out-of -scale.

2) what are the differences between PANC-1 and MIAPaCa-2 cell lines?

Both cell lines derived from patients with PDAC and shows similar mutation profile, as described in Moore P.S. et al Genetic profile of 22 pancreatic carcinoma cell lines Analysis of K-ras, p53, p16 and DPC4/Smad4. Virchows Arch (2001) 439:798–802 DOI 10.1007/s004280100474

3) English language should be edited by a native English speaker

English was revised as requested.

4) All non-standard abbreviations should be defined the first time they are used

Abbreviations were defined as requested.

Reviewer 5 Report

Title: Evening Primrose Oil improves chemotherapeutic effects in Human Pancreatic Ductal Adenocarcinoma cell lines. A preclinical study.

Although the study looks interesting there are some minor correction with the following findings:

  1.  Introduction: Author can improve this section by including the general overview on cancer in brief before introducing the pancreatic cancer.
  2. Discussion: a.  Authors have to include the limitation of their findings in this section. b. Authors also have to elaborate future prospective  of the current findings in this section.
  3. Overall findings looks promising.

Author Response

As requested, we responded to the reviewer’s comments and the new data were added to the manuscript.

Review 4 response

1)         panel B in fig. 6, axes and points should be labeled (e.g., which doses correspond to point 1? and so on; there are less than 9 points on the graphs, where are the missing points?), and the authors should comment the results of the analysis.

Thank you for these appropriate comments. The Figure 6 was modified as requested, and result comments described. Regarding the 9 points, the software CompuSyn generates the isobologram excluding the C.I. values out-of -scale.

2) what are the differences between PANC-1 and MIAPaCa-2 cell lines?

Both cell lines derived from patients with PDAC and shows similar mutation profile, as described in Moore P.S. et al Genetic profile of 22 pancreatic carcinoma cell lines Analysis of K-ras, p53, p16 and DPC4/Smad4. Virchows Arch (2001) 439:798–802 DOI 10.1007/s004280100474

3) English language should be edited by a native English speaker

English was revised as requested.

4) All non-standard abbreviations should be defined the first time they are used

Abbreviations were defined as requested.

Reviewer 5 response:

Brief introduction of a general overview in cancer was added as well as future perspectives and limitation of these data.

Round 2

Reviewer 2 Report

Satisfactory changes were done and still minor corrections are required regarding the spelling check.

Author Response

Thanks for the revision, English revision was performed.

Reviewer 3 Report

EPO are synergistic with gemcitabine for Pancreatic ductal Adenocarcinoma application. Authors used HaCaT cell line as normal which is not persuasive (Figure 1B).

Author Response

EPO is synergistic with Pacliitaxel and not with Gemcitabine as described in fig. 6B.

Thanks for the revision.